

# Comprehensive analysis of bioplastics: life cycle assessment, waste management, biodiversity impact, and sustainable mitigation strategies

Kushi Yadav[1] and Ganesh Chandrakant Nikalje[2]

[1] Amity Institute of Biotechnology, Amity University, Noida, Uttar Pradesh, India
[2] Department of Botany, Seva Sadan's R. K. Talreja College of Arts, Science and Commerce, University of Mumbai, Ulhasnagar, India

## ABSTRACT

Bioplastics are emerging as a promising alternative to traditional plastics, driven by the need for more sustainable options. This review article offers an in-depth analysis of the entire life cycle of bioplastics, from raw material cultivation to manufacturing and disposal, with a focus on environmental impacts at each stage. It emphasizes the significance of adopting sustainable agricultural practices and selecting appropriate feedstock to improve environmental outcomes. The review highlights the detrimental effects of unsustainable farming methods, such as pesticide use and deforestation, which can lead to soil erosion, water pollution, habitat destruction, and increased greenhouse gas emissions. To address these challenges, the article advocates for the use of efficient extraction techniques and renewable energy sources, prioritizing environmental considerations throughout the production process. Furthermore, the methods for reducing energy consumption, water usage, and chemical inputs during manufacturing by implementing eco-friendly technologies. It stresses the importance of developing robust disposal systems for biodegradable materials and supports recycling initiatives to minimize the need for new resources. The holistic approach to sustainability, including responsible feedstock cultivation, efficient production practices, and effective end-of-life management. It underscores the need to evaluate the potential of bioplastics to reduce plastic pollution, considering technological advancements, infrastructure development, and increased consumer awareness. Future research should focus on enhancing production sustainability, understanding long-term ecological impacts, and advancing bioplastics technology for better performance and environmental compatibility. This comprehensive analysis of bioplastics' ecological footprint highlights the urgent need for sustainable solutions in plastic production.

# INTRODUCTION

Bioplastics are materials made from renewable sources like plants, biomass, or microorganisms, offering a more sustainable alternative to traditional plastics (*Rosenboom, Langer & Traverso, 2022*). They can be bio-based, biodegradable, or both

Corresponding author
Ganesh Chandrakant Nikalje, gcnikalje@gmail.com

(*Cywar et al., 2022*). The goal of bioplastics is to address the environmental issues linked with conventional plastics, which are made from fossil fuels and have harmful effects throughout their life cycle, from extraction and production to use and disposal (*Atiwesh et al., 2021*). Bioplastics aim to reduce environmental impact by offering several benefits: lower carbon footprints, better resource conservation, less waste, and support for a circular economy (*Degli Esposti et al., 2021*). Bio-based bioplastics often use crops or forestry by-products, which helps cut carbon emissions and reduces reliance on fossil fuels (*Mongkoldhumrongkul & Sukkanta, 2022*; *Rosenboom, Langer & Traverso, 2022*). They also offer a way to use fewer finite resources, such as petroleum, which is the base material for traditional plastics (*Zhao et al., 2023*). Bioplastics can potentially improve waste management. Being biodegradable, they break down into natural elements, reducing plastic waste in landfills and oceans (*Folino et al., 2020*). Some can be recycled or composted, supporting circular economy practices by promoting reuse and recycling. With an annual production of around two million tonnes, 100% bio-based bioplastics are seen as crucial for future circular economies and for achieving United Nations Sustainable Development Goals (SDGs) (*Kumar et al., 2021*). These goals include using less toxic substances, developing new recycling methods, and reducing dependence on fossil fuels (*Rosenboom, Langer & Traverso, 2022*). Bioplastics generally have a lower environmental impact than traditional plastics and offer sustainable alternatives (*Altalhi, 2022*). Their adoption drives research and innovation in sustainable materials (*Palsra & Chauhan, 2020*). This review examines various methods for assessing bioplastics' popularity and adoption, including life cycle assessments from production to disposal. It also explores production methods, advancements, waste management strategies, and their effects on biodiversity and ecosystems. The review concludes with practices to reduce environmental impact and promote the sustainable use of bioplastics. Ongoing advancements in bioplastics technology continue to improve their performance, durability, and affordability, increasing their potential for widespread use.

## SURVEY METHODOLOGY

The study used a detailed questionnaire to explore awareness and perceptions of bioplastics. The sample included people from 42 countries, mainly in Europe and Asia (*Filho et al., 2022*). The survey was designed to gather a range of insights through self-directed responses. It covered demographics, knowledge of bioplastics, consumption habits, concerns, and opinions. The questionnaire used various answer formats, including multiple-choice and frequency scales, and was carefully reviewed for clarity and relevance. The survey was conducted online *via* the LimeSurvey platform over 5 months, gathering 384 responses. The data was analyzed using SPSS V.26, with frequency analyses and Chi-square tests examining the relationships between education, age, and gender, using a significance level of 5% (*Filho et al., 2022*).

Our research aimed to understand the environmental impact of bioplastics, motivated by the growing importance of sustainable materials in addressing environmental issues. We used a carefully designed survey to gather detailed insights from consumers, industry

professionals, and environmental experts. A preliminary test of the survey with a small group was conducted to refine the questions and ensure clarity and objectivity (*Bishop, Styles & Lens, 2021*). To investigate the environmental effects of bioplastics, we performed an extensive literature review using the PubMed database. We started with search terms like "bioplastics environmental impact" and "sustainable materials footprint" and refined our search to include terms such as "biodegradable plastics life cycle assessment" and "renewable materials carbon footprint." We also explored terms like "bioplastic manufacturing environmental effects" and "biodegradable polymer waste management." This thorough search strategy aimed to provide a comprehensive understanding of the ecological impacts of bioplastics, forming the basis for our research into their environmental footprint. The data collection process involved a detailed review of recent publications and relevant literature. This careful approach ensured data accuracy and adherence to established methods. The collected data was thoroughly analysed to provide a clear and meaningful understanding of various research findings on bioplastics. Ethical considerations were crucial throughout the survey process. The review clearly explained the study's purpose and maintained transparency. In summary, this survey methodology was carefully crafted to offer a thorough and reliable examination of the ecological impact of bioplastics. By systematically gathering diverse perspectives and maintaining high ethical standards, the study aimed to provide valuable insights into sustainable materials and environmental conservation. The goal was to significantly advance understanding of bioplastics' environmental effects and support the development of sustainable practices and policies.

## EMERGING TRENDS IN BIOPLASTIC ADOPTION

In recent years, bioplastics have gained popularity as a sustainable alternative to traditional plastics. Growing global concern about environmental issues has increased interest in bioplastics among individuals, businesses, and governments (*Moshood et al., 2021*). Bioplastics are favoured for their reduced carbon footprint, renewable sources, and potential biodegradability (*Coppola et al., 2021*, Fig. S1). Governments around the world are enacting regulations to promote sustainable materials and reduce plastic waste through bans, taxes, and incentives (*Adam et al., 2020*; *Kiessling et al., 2023*). These policies support the growth of bioplastics. Companies are incorporating sustainability into their strategies, using bioplastics in packaging, manufacturing, and product innovation to meet the rising consumer demand for eco-friendly alternatives (*Ashrafi et al., 2019*; *Westlake et al., 2023*). Consumers increasingly prefer brands that use bioplastics, seeing them as responsible and sustainable options (*Filho et al., 2022*; *Galati et al., 2022*). Technological advancements have improved bioplastic manufacturing, research, and material development (*Andreeßen & Steinbüchel, 2019*). These innovations have made bioplastics more appealing across various industries. Collaboration between governments, industry, and NGOs has driven progress and created sustainable supply chains (*Maione, Lapko & Trucco, 2022*). Increased production has made bioplastics more accessible and affordable (*Gong et al., 2023*). The rise of bioplastics is supported by environmental awareness, regulations, corporate

sustainability, consumer demand, technological advancements, and collaborative efforts (*Wydra et al., 2021*), ensuring their continued growth as a sustainable solution. Evaluating the environmental impact of bioplastics throughout their life cycle—from production to disposal—is essential. While bioplastics are often seen as more eco-friendly than traditional plastics, it is important to assess their specific environmental effects (*Atiwesh et al., 2021*). Key concerns include the environmental footprint of feedstock production. Large-scale monoculture farming can lead to deforestation, habitat loss, soil erosion, water pollution from pesticides and fertilizers, and depletion of water resources (*Shaheen & Sabir, 2017*; *Van Roijen & Miller, 2022*). The production of bioplastics involves energy-intensive processes like fermentation, chemical synthesis, and polymerization. The greenhouse gas (GHG) emissions from these processes can vary based on the feedstock and production methods used (*Chong et al., 2022*; *Jin et al., 2023*). The impact is also affected by the type of energy used in manufacturing.

Addressing excessive carbon dioxide emissions, which contribute to global warming, highlights the need for improved methods of converting carbon dioxide into useful products such as biomethane, bioethanol, polyhydroxybutyrate, and succinic acid. Microorganisms like bacteria, algae, and yeast play a crucial role in this process. Advances in genetic engineering and machine learning are improving the efficiency of capturing and converting carbon dioxide through various methods including photoautotrophic biosynthesis, dark fermentation, and biodegradation. For instance, *Chlorella vulgaris* and Cyanobacteria can capture over 90% of carbon dioxide and produce approximately 0.45 g/L/day of biomass in a standard photobioreactor (*Akash et al., 2023*). The production of bioplastics can require significant amounts of water for feedstock irrigation, processing, and cooling (*Morão & De Bie, 2019*). The impact on local water resources depends on regional water availability, so it is important to manage water use sustainably. Another concern is the competition for feedstock crops, which could affect food security and land use if these crops are grown instead of food crops (*Bishop, Styles & Lens, 2022*). The water footprint of bioplastics ranges from 1.4 to 9.5 cubic meters per kilogram, while the land footprint ranges from 0.7 to 13.75 square meters per kilogram. If bioplastics were to replace all fossil-based plastics, the annual water footprint could range from 307 billion to 1,652 billion cubic meters, representing 3% to 18% of the global annual average. Similarly, the land footprint could range from 30 million to 219 million hectares per year, potentially using 8% to 61% of available arable land (*Putri, 2018*). To reduce environmental impact and ensure responsible resource use, it is crucial to adopt sustainable sourcing and land management practices for bioplastics production. Effective waste management is also important for biodegradable bioplastics to fully realize their environmental benefits. In landfills without proper conditions for degradation, biodegradable bioplastics can release methane, a potent greenhouse gas (*Atiwesh et al., 2021*). If bioplastics are not sorted from conventional plastics, they can contaminate recycling streams and reduce the quality of recycled materials. Poor waste disposal can lead to bioplastics entering natural ecosystems like oceans and rivers, where they can cause pollution and harm wildlife through ingestion and entanglement (*Hahladakis, 2020*).

## LCA OF BIOPLASTICS

The Life Cycle Assessment (LCA) is a detailed process used to evaluate the environmental impact of bioplastic products throughout their entire life cycle, from raw material extraction to manufacturing, use, and end-of-life (EOL) management (*Gadaleta et al., 2023*). This comprehensive analysis covers crucial stages, including raw material sourcing, production, use, and disposal (*Ali et al., 2023*). The LCA process begins with clearly defining the assessment's objectives and scope. This includes specifying the goals, boundaries, functional units (such as per kilogram of bioplastic), and system limits (*Salwa et al., 2021*; *Alhazmi, Almansour & Aldhafeeri, 2021*). Once these parameters are established, the Life Cycle Impact Assessment (LCIA) is conducted, where the collected Life Cycle Inventory (LCI) data is analyzed to assess the potential environmental impacts at various life cycle stages. This analysis uses different methodological frameworks outlined in scholarly research (*Beckstrom et al., 2020*; *Chalermthai et al., 2021*). Identifying environmental hotspots and areas of significant impact through this rigorous analysis enables informed decision-making and strategic actions to reduce environmental footprints (*Talwar & Holden, 2022*). The insights gained from LCA can lead to various interventions, such as optimizing processes, integrating renewable energy sources, minimizing waste, and enhancing recycling infrastructure (*Ali et al., 2023*; *Kakadellis & Harris, 2020*). Moreover, sharing LCA results through thorough reporting and verification, in line with standards like ISO 14040 and ISO 14044, strengthens the credibility and reliability of the findings. There is also an option for additional third-party validation to ensure compliance with recognized academic standards (*Di Bartolo, Infurna & Dintcheva, 2021*).

Comparative studies of the carbon, environmental, and water footprints between conventional polypropylene (PP) plastic and bioplastic fibres have shown significant benefits for bioplastics (*Broeren et al., 2017*). Specifically, bioplastic fibres have a lower carbon footprint and overall reduced environmental impact compared to PP. Additionally, incorporating starch in the production of biodegradable bioplastics has led to notable reductions in greenhouse gas (GHG) emissions and non-renewable energy use, highlighting the potential for environmental benefits in bioplastic production (*Ali et al., 2023*). However, using starch in bioplastics can increase eutrophication potential and land use compared to petrochemical plastics, indicating the trade-offs in formulation (*Ali et al., 2023*). Incorporating residual starch residues in bioplastic blends shows promise in mitigating these negative impacts, leading to reductions in land use, eutrophication potential, GHG emissions, and non-renewable energy consumption (*Ali et al., 2023*). Additionally, strategies to reduce the water footprint, such as using residual vegetative biomass from various crop sources, offer further opportunities for sustainable production (*Ali et al., 2023*). Despite these environmental benefits, challenges remain in the commercial viability of bioplastics, particularly polyhydroxyalkanoates (PHA), due to their higher production costs compared to fossil fuel-derived plastics (*Khatami et al., 2021*). Efforts to lower these costs have been hindered by factors such as slow microbial growth, inefficient raw material conversion, high energy demands, and expensive downstream

processing (*Mannina et al., 2020*). The study combined the carbon, environmental, and water footprints of regular PP plastic with those of bioplastic fibres. The results showed that bioplastic fibres have a smaller carbon footprint and a lower overall environmental impact compared to PP (*Ali et al., 2023*). Using starch in the production of biodegradable bioplastics reduced greenhouse gas (GHG) emissions by up to 80% and non-renewable energy use by up to 60% (*Broeren et al., 2017*). However, compared to petrochemical plastics, starch can increase eutrophication potential by up to 400% and land use by 0.3 to 1.3 square meters per kilogram of bioplastic (*Broeren et al., 2017*). Blending starch with residual starch residues can help reduce these impacts, lowering land use by up to 60%, eutrophication potential by up to 40%, GHG emissions by up to 10%, and non-renewable energy use by up to 60%. Additionally, using residual vegetative biomass from various crops can help reduce the water footprint (*Broeren et al., 2017*). Despite these environmental benefits, the commercial production of PHA bioplastics remains challenging due to higher costs compared to fossil fuel-based plastics. In 1998, PHA was up to 1,700% more expensive than fossil-based plastic, and while the price has since dropped to around €5 per kilogram, it is still higher than synthetic plastic, which costs between €0.80 and €1.50 per kilogram (*Khatami et al., 2021*). The high cost of PHA production is due to slow microbial growth, inefficient raw material conversion, high energy requirements for sterilization and aeration, and expensive downstream processing (*Mannina et al., 2020*).

The environmental performance of bioplastics can vary depending on factors like feedstock type, farming methods, energy sources, and EOL management options (*Benavides, Lee & Zarè-Mehrjerdi, 2020*). Additionally, LCA studies should consider not just environmental impacts but also social and economic factors to provide a more comprehensive assessment of sustainability (*Bishop, Styles & Lens, 2021*). When comparing the LCA of bioplastics to traditional plastics, it is crucial to consider that bioplastics often rely on renewable resources like plant-based feedstock (*e.g.*, corn, sugarcane, or cellulose), while traditional plastics typically use fossil fuel-based feedstock such as petroleum or natural gas (*Muthusamy & Pramasivam, 2019*). Bioplastics generally have a lower environmental impact during the extraction phase due to their renewable nature. Their manufacturing processes often require less energy compared to conventional plastics, especially when efficient technologies and renewable energy sources are used. In contrast, traditional plastics involve energy-intensive processes like polymerization and refining crude oil, leading to higher carbon emissions and other environmental impacts (*Rosenboom, Langer & Traverso, 2022*). Bioplastics also have the potential to offer more sustainable EOL solutions than standard plastics (*Paul-Pont et al., 2023*). Some are designed to be compostable, breaking down into organic matter under specific conditions (*Gioia et al., 2021*). However, it is important to note that not all bioplastics are biodegradable, and proper composting infrastructure is required for those that are. Traditional plastics, unless recycled, often end up in landfills or incineration, causing long-term environmental issues (*Zhu & Wang, 2020*).

Traditional plastics like polyethylene (PE) and PP have well-established recycling systems and can be recycled multiple times (*Fredi & Dorigato, 2021*). For example,

reprocessed PP from meat trays showed reduced Melt Flow Index (MFI) values, indicating it was initially produced through extrusion and thermoforming processes. However, slight increases in MFI during reprocessing suggest some degradation, making it unsuitable for closed-loop recycling into new trays. Additionally, plastic strings from these samples had rough surfaces, differing from other samples (*Eriksen et al., 2019*). A study found that reprocessed PE from soap bottles maintained consistent MFI values, indicating limited degradation and suitability for closed-loop recycling into new bottles (*Barletta et al., 2019*). The European Union (EU) has implemented several policies under the European Green Deal and its Circular Economy Action Plan. These include a recycling target of 50% for plastic packaging by 2030 and a ban on various single-use plastic items, such as polystyrene straws, cutlery, food containers, and oxo-degradable plastics, effective from January 2021 (*Hoang et al., 2022*; *Rosenboom, Langer & Traverso, 2022*). In contrast, the recycling infrastructure for bioplastics is less developed and varies depending on the type of bioplastic. Improperly sorted bioplastics can contaminate the recycling stream, and their degradability in natural environments depends on their composition and conditions, with some requiring industrial composting facilities to degrade fully (*Folino et al., 2020*). LCAs comparing bioplastics and traditional plastics have shown varied results depending on the materials and systems studied. In some cases, bioplastics have demonstrated lower carbon emissions (7.60–73.75% lower) and reduced environmental impacts, especially when renewable energy sources are used during production (*Chen et al., 2024*).

In 2020, *Benavides, Lee & Zarè-Mehrjerdi (2020)* conducted a comprehensive LCA to examine the environmental impact of biodegradable polylactic acid (PLA) and bio-based polyethylene (bio-PE) plastics, from raw material sourcing to the EOL phase. The study compared these bioplastics to traditional fossil-based plastics like high-density polyethylene (HDPE) and low-density polyethylene (LDPE). The results showed that bio-PE and PLA had lower greenhouse gas (GHG) emissions, with minimal levels recorded at 1.0 and 1.7 kg $CO_2$ equivalent per kg, respectively, when biodegradation did not occur. Fossil energy consumption (FEC) was also lower for these bio-derived plastics, with bio-PE using 29 and PLA 46 MJ per kg. However, the study highlighted a significant issue with PLA's environmental performance when it biodegrades in landfills and composting environments. In these cases, PLA's life cycle emissions increased significantly, by 16% to 163%, compared to scenarios where it did not degrade in landfills. This study provides valuable insights into how biodegradability can influence GHG emissions in landfill settings. However, other environmental factors like land and water use, resource depletion, and potential impacts on biodiversity should also be considered, as they may vary depending on the specific context (*Atiwesh et al., 2021*). The comparison of LCA related parameters of bioplastics and conventional plastics are tabulated in Table 1. When considering key environmental factors, energy consumption is a major concern. Bioplastics typically require less energy to produce compared to traditional plastics, though this can vary depending on the type of bioplastic and manufacturing process (*Bishop, Styles & Lens, 2021*). Some bioplastics are produced using renewable energy, which further reduces their carbon footprint (*Rosenboom, Langer & Traverso, 2022*). However, it is important to account for the energy used in growing raw materials, transportation, and

**Table 1 Comparative account of life cycle assessment related parameters of bioplastics and conventional plastics.**

| Sr. no. | Parameter | Bioplastic | Conventional plastics |
|---|---|---|---|
| 1 | Raw material extraction | Made from renewable biomass sources like corn starch, sugarcane, or cellulose. The extraction process generally has a lower carbon footprint compared to fossil fuel extraction. | Derived from petroleum, requiring energy-intensive extraction and refining processes with significant greenhouse gas emissions. |
| 2 | Energy consumption | Production generally consumes less energy due to the use of renewable sources. However, the total energy required can vary based on the type of bioplastic and agricultural practices. | Typically, higher energy consumption due to the extraction, refining, and polymerization of fossil fuels. |
| 3 | Greenhouse gas emissions | Generally, emit fewer greenhouse gases during production. However, emissions can vary significantly based on the feedstock and production methods. | Higher greenhouse gas emissions due to fossil fuel use and energy-intensive production processes. |
| 4 | Land use | Require agricultural land for growing feedstock, which can lead to land-use changes, deforestation, and competition with food production. | Do not require agricultural land, but their extraction and refining processes can cause environmental degradation. |
| 5 | Water use | Production can be water-intensive, especially for irrigation of crops used as feedstock. This varies based on agricultural practices and regional water availability. | Water use is generally lower than bioplastics but still significant in extraction and refining processes. |
| 6 | Resource depletion | Utilize renewable resources, potentially reducing the depletion of finite fossil fuel reserves. However, the sustainability of biomass feedstocks can be an issue if not managed properly. | Depend on non-renewable fossil fuels, contributing to resource depletion. |
| 7 | Biodegradability and end-of-life | Many bioplastics are designed to be biodegradable or compostable, reducing their impact on landfills and marine environments. The actual biodegradability depends on specific conditions (*e.g.*, industrial composting *vs.* home composting). | Generally, not biodegradable, leading to long-term environmental pollution, especially in marine ecosystems. |
| 8 | Toxicity and pollution | Lower risk of releasing toxic substances during degradation, although the use of fertilizers and pesticides in agriculture can cause pollution. | Can release harmful chemicals during production, use, and disposal. Microplastic pollution is a major environmental concern. |
| 9 | Economic factors | Typically, more expensive due to current production scales and technology. However, costs are decreasing as technology improves and production scales up. | Generally cheaper due to established production processes and economies of scale. |
| 10 | Social impact | Potential to create new agricultural and manufacturing jobs, but may also lead to food *vs.* fuel debates and land-use conflicts. | Established industry with significant economic impact, but associated with negative health impacts in communities near extraction and refining sites. |

**Note:**
References: (*Hobbs et al., 2021*; *Bishop, Styles & Lens, 2021*; *Samanta et al., 2022*; *Ali et al., 2023*).

end-of-life management when evaluating overall energy consumption. Bioplastics, like corn-based PLA, can reduce greenhouse gas (GHG) emissions by up to 25% compared to standard plastics (*Atiwesh et al., 2021*). The Biopolymer Feedstock System (BPFS) offers significant environmental benefits, potentially replacing plastics like PP and nylon, which have higher environmental impacts. For example, PP and nylon have carbon footprints of 1.98 and 8.03 kg $CO_2$eq/kg, respectively (*Wernet et al., 2016*). In a conservative scenario, BPFS can reduce emissions by 67% compared to traditional plastics, with emissions as low as 0.656 kg $CO_2$eq/kg BPFS. In the best-case scenario, emissions reductions can reach up to 116% (*Crocker et al., 2020*). These findings highlight the environmental advantages of

adopting BPFS to reduce plastic-related emissions. During production, bioplastics often generate fewer emissions due to the use of renewable feedstock and more energy-efficient processes. However, total GHG emissions depend on factors such as feedstock type, cultivation practices, manufacturing methods, and end-of-life options (*Coppola et al., 2021*). It is crucial to consider emissions from all life cycle stages, including the effects of land use changes on carbon sequestration. Water usage in bioplastic production varies depending on the feedstock and manufacturing process (*Ita-Nagy et al., 2020*). For example, bioplastics made from water-intensive crops like sugarcane may require more water during cultivation. However, compared to traditional plastics, bioplastics generally use less water during the manufacturing stage (*Samberger, 2022*). When assessing the environmental impact of bioplastics, it is important to consider both direct water use and potential effects on local water resources, such as pollution or water scarcity. Land and resource requirements for bioplastics depend on the feedstock and cultivation methods used (*Chaplin-Kramer et al., 2017*). Bioplastics made from agricultural crops require land, which can lead to land-use changes and competition with food production. However, some bioplastics are made from non-food sources, agricultural waste, or algae, reducing land use conflicts (*Karan et al., 2019*). Sustainable land management practices, such as regenerative agriculture, can help mitigate environmental impacts. Resource utilization should also consider feedstock availability, extraction methods, and potential effects on biodiversity and ecosystem services (*Schulte et al., 2022*).

## OVERVIEW OF BIOPLASTIC PRODUCTION METHODS

Bioplastic production involves various methods that use renewable resources to create plastic-like materials. Two prominent methods are fermentation and microorganism-based processes. For instance, PLA is made through the fermentation of renewable feedstocks. Microorganisms like bacteria or yeast convert sugars from plant sources, such as corn or sugarcane, into lactic acid, which is then used to produce PLA (*Reshmy et al., 2021*). Another method involves using starch from crops like corn, wheat, or potatoes. This process, known as starch blending, mixes starch with plasticizers and additives to improve its properties. The resulting blend can be processed using traditional plastic manufacturing techniques, such as extrusion, injection molding, or blow molding (*Carneiro da Silva, Rios & Campomanes Santana, 2023*; *Dewi et al., 2023*). Some bioplastics are created through chemical processes with renewable raw materials. For example, PHAs are biodegradable bioplastics produced by fermenting plant-based sugars or lipids. These polymers can be used to make various plastic products (*Francis & Parayil, 2023*). Bio-PE is another type of bioplastic made by partially replacing fossil fuel-derived ethylene with ethylene from renewable sources like sugarcane or ethanol. Although it follows the same polymerization process as traditional PE, Bio-PE has a lower carbon footprint because of its renewable feedstock (*Zhao et al., 2023*). The comparative analysis of bioplastic production methods is shown in Table 2.

Algae are promising feedstocks for bioplastics due to their fast growth and ability to absorb carbon dioxide. Key biopolymer precursors from algae, such as alginate and

**Table 2 Comparative analysis of Bioplastic production methods.**

| Sr. No. | Method | Source | Yield | Challenge | Reference |
|---|---|---|---|---|---|
| 1 | Fermentation | Sugars, starches, oils | Moderate | High production costs, scale-up challenges, environmental impact of feedstock cultivation | *Bhatia et al. (2021)* |
| 2 | Chemical synthesis | Biomass, fossil fuels | Moderate | Dependency on fossil fuels, energy-intensive processes, generation of by-products, limited biodegradability | *Coppola et al. (2021)* |
| 3 | Bio-based polymer blends | Various biobased sources | High | Compatibility issues between different polymers, performance variability, recycling complexities, environmental impact of additives | *Censi et al. (2022)* |
| 4 | Enzymatic polymerization | Monomers, polymers | High | Limited substrate specificity, control over polymerization parameters, enzyme stability and recyclability | *Fernandes et al. (2022)* |

carrageenan, are used to make algae-based bioplastics. Research is focused on improving production processes and material properties (*Cheah et al., 2023*). Another approach is blending bio-based polymers with petroleum-derived plastics, like PLA with PET, to enhance performance and reduce the overall carbon footprint (*Gironi & Piemonte, 2011*; *Mori, 2023*). The environmental impact of bioplastic production depends largely on how feedstocks are cultivated, including land use, water consumption, and biodiversity preservation (*Thomas et al., 2023*). Unsustainable practices, like excessive pesticide use or deforestation, can lead to soil erosion, water contamination, habitat loss, and greenhouse gas emissions (*Ortas, 2023*). Therefore, choosing the right feedstock and practicing sustainable cultivation are crucial for sustainability. The methods used to extract raw materials, whether through fermentation or chemical synthesis, also affect energy consumption, water use, emissions, and waste (*Ortega et al., 2022*). To reduce these impacts, it is important to use renewable energy, efficient extraction technologies, and optimize resource use (*Luderer et al., 2022*). The manufacturing of bioplastics involves energy-intensive processes, such as polymerization, compounding, and shaping, which contribute to greenhouse gas emissions and environmental pollution (*Atiwesh et al., 2021*). To minimize these impacts, adopting energy-efficient equipment, improving production techniques, and using cleaner technologies are essential steps. The disposal of standard plastics and bioplastics differs significantly in terms of environmental impact and sustainability. Traditional plastics are usually disposed of through landfilling or incineration, which can cause long-term pollution and environmental harm (*Bishop, Styles & Lens, 2021*). In contrast, bioplastics, especially those designed to be biodegradable or compostable, offer more environmentally friendly disposal options. Biodegradable bioplastics break down into natural substances, reducing plastic waste and its negative effects. Compostable bioplastics can be added to organic waste streams and turned into nutrient-rich compost (*Mhaddolkar et al., 2024*). Overall, bioplastic disposal can be a more sustainable choice compared to traditional plastics, potentially lowering plastic waste and environmental impact. However, effective waste management practices are crucial to fully benefit from these environmental advantages.

# BIOPLASTIC WASTE MANAGEMENT

## Degradation and disposal techniques of bioplastics

When considering the EOL stage for bioplastics, the focus is on how they degrade and are disposed of. Composting is a common method for bioplastics made from materials like starch or PLA, which are specifically designed to be compostable (*Cucina et al., 2021*). During composting, organic matter is broken down into nutrient-rich compost in a controlled environment. Certified compostable bioplastics can decompose in industrial composting facilities, where conditions like temperature, humidity, and microbial activity help them break down into water, carbon dioxide, and biomass within a set period (*Nazaruddin & Jamari, 2021*). Composting is a sustainable disposal method as it creates valuable compost that enriches the soil (*Al-Alawi et al., 2019*). Another method is anaerobic digestion, where microorganisms break down organic materials without oxygen. Some bioplastics, such as certain biopolyesters, can be processed this way to produce biogas (*Benyahya et al., 2021*).

The disposal of bioplastics involves several methods, each with its own environmental implications. Anaerobic digestion uses microorganisms to break down bioplastics without oxygen, producing biogas (mainly methane and carbon dioxide) for energy (*Coltelli et al., 2021*). Mechanical recycling involves sorting, cleaning, shredding, and reprocessing bioplastic waste into granules or pellets, which helps reduce the need for new raw materials. Its effectiveness depends on the type of bioplastic, contamination levels, and recycling infrastructure (*Al-Qadri et al., 2022*). Chemical recycling breaks bioplastics down into molecular components, which can be turned into new products. Although it can handle mixed plastics, it requires specialized facilities and ongoing development to be economically viable (*Agüero et al., 2023*). Incineration involves burning bioplastics in waste-to-energy facilities, generating heat. Bioplastics produce about 17 to 25 MJ/kg of heat energy on average. However, this method also releases carbon dioxide and other pollutants, such as volatile organic compounds (VOCs), nitrogen oxides (NOx), and particulate matter, which can harm health and the environment (*Kalair et al., 2021*). Composting is another method, particularly for bioplastics designed to be compostable. It turns bioplastics into organic matter in composting facilities, reducing carbon emissions compared to traditional composting processes (*Vinci et al., 2021*). The choice of disposal method for bioplastics depends on factors like the specific type of bioplastic, local waste management infrastructure, and environmental considerations (*Khodaei, Álvarez & Mullen, 2021*). Effective management also requires proper collection systems, clear labelling, and consumer education.

## Comparison of biodegradable, compostable, and non-biodegradable bioplastics

Bioplastics can be categorized into three main types based on their ability to break down and return to the environment: biodegradable, compostable, and non-biodegradable. Biodegradable bioplastics are designed to decompose naturally into simpler substances such as water, carbon dioxide, and biomass through biological processes. This

decomposition is aided by microorganisms like bacteria or fungi and can occur in various environments, including soil, water, or industrial composting facilities. However, biodegradability does not guarantee a specific timeframe for decomposition or complete breakdown. If managed properly, biodegradable bioplastics can help reduce waste and minimize environmental impacts (*Mendieta et al., 2022*; *Kochanska et al., 2022*; *Dzeikala et al., 2023*). Compostable bioplastics are a subset of biodegradable plastics that meet specific criteria for breaking down in industrial composting facilities (*Mhaddolkar, Koinig & Vollprecht, 2022*). Standards such as ASTM D6400 or EN 13432 define these criteria, which include specific temperature, humidity, and microbial conditions. Compostable bioplastics must be disposed of in industrial composting facilities rather than backyard composting to ensure proper degradation. The resulting compost can then be used to enrich soil (*Santi, Elegir & Del Curto, 2020*; *Dzeikala et al., 2023*). Non-biodegradable bioplastics do not easily decompose under normal environmental conditions and often have similar properties to traditional plastics, such as durability and longevity. These bioplastics resist natural breakdown and may need to be managed through recycling or energy recovery methods to mitigate their environmental impact (*Reichert et al., 2020*; *Siddiqui et al., 2023*). Choosing between biodegradable, compostable, or non-biodegradable bioplastics depends on application needs, waste management capabilities, and environmental considerations (*Vaverková et al., 2012*; *Falzarano et al., 2023*).

## Enhancing sustainability: disposal impact and recycling ventures
### Environmental consequences of improper disposal and lack of infrastructure

Inadequate disposal practices and insufficient infrastructure for bioplastics can lead to significant environmental issues. Improperly discarded bioplastics in oceans, rivers, or forests can persist for long periods, harming wildlife through entanglement or ingestion. This pollution disrupts ecosystems, threatens biodiversity, and damages habitats (*Clause, Celestian & Pauly, 2021*). In landfills, bioplastics not designed for biodegradation can release harmful chemicals as they break down, potentially contaminating soil and groundwater, which affects local ecosystems and human health (*Okoffo et al., 2022*). When bioplastics are disposed of in landfills or incinerated instead of being composted or recycled, they can contribute to greenhouse gas (GHG) emissions, reducing their intended environmental benefits (*Maione, Lapko & Trucco, 2022*). A lack of infrastructure and appropriate waste management systems often results in bioplastics ending up in mixed waste streams or conventional recycling systems where they may not be properly sorted or recycled. This results in missed opportunities for recycling or composting, increases waste generation, and depletes resources (*Kovačević, Flinčec Grgac & Bischof, 2021*). Bioplastics, which are largely made from renewable resources like plant-based feedstocks, lose their potential for recovery and reuse when not managed correctly. This worsens reliance on fossil fuel-based plastics and contributes to resource depletion. To address these challenges, it is crucial to develop and implement robust waste management infrastructure, including composting and recycling facilities, along with clear labelling and disposal guidelines for bioplastics (*Islam & Cullen, 2023*). Education and awareness campaigns are

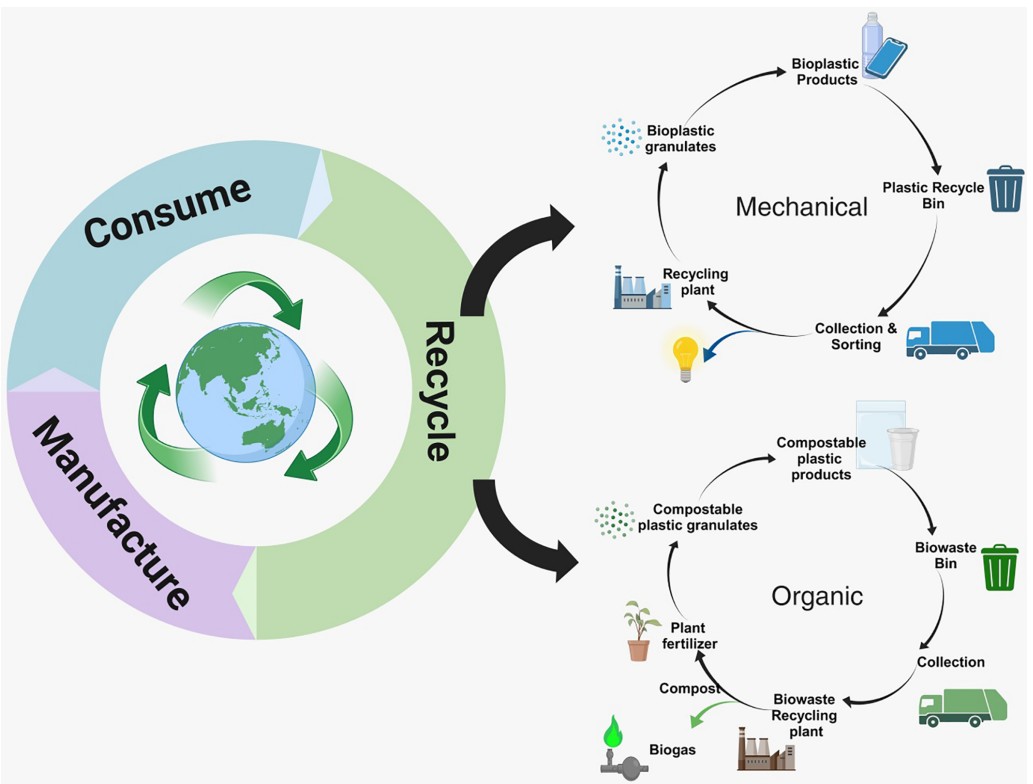

**Figure 1 Circular life cycle and recycling pathways of plastic products.** The circular life cycle of plastic products, highlights the stage of manufacture, consumption, and recycling. It shows two primary recycling pathways: mechanical recycling, organic recycling, where biodegradable plastics are composted and returned to the ecosystem. (Modified from *Ali et al. (2023)*).

also essential to encourage responsible consumer behaviour and promote sustainable disposal practices. Effective management of bioplastics requires collaboration among governments, industry stakeholders, and consumers to minimize environmental impacts and leverage bioplastics as a sustainable alternative to traditional plastics (*Moshood et al., 2022*).

### *Exploration of recycling and composting options for bioplastics*

Recycling and composting are important options for managing bioplastics at the end of their life cycle.

#### *Recycling*

Mechanical recycling involves several steps to manage bioplastic waste. The process starts with collecting and sorting the waste, followed by cleaning and shredding it. The shredded material is then melted and formed into pellets or flakes, which can be used as raw materials for making new bioplastic products. This closed-loop recycling method helps reduce the need for virgin materials and minimizes waste, as shown in Fig. 1 (*Cabrera et al., 2022*). However, the effectiveness of mechanical recycling depends on factors such as the type of bioplastic, contamination levels, and the availability of recycling facilities.
Chemical recycling, also known as feedstock or depolymerization recycling, breaks down bioplastics into their basic molecular components using chemical processes. This method converts bioplastics into monomers or other valuable chemicals, which can be used to produce new plastics or other products (*Agüero et al., 2023*). Chemical recycling is particularly useful for handling mixed or complex bioplastic materials that are not suitable for mechanical recycling. However, this technology is still developing and requires specialized facilities and further improvements to become more scalable and economically viable (*Agüero et al., 2023*).

*Composting*

Composting is a controlled process that breaks down organic waste, including some bioplastics, into nutrient-rich compost (*Sayara et al., 2020*). Industrial composting facilities create the ideal conditions—such as temperature, humidity, and microbial activity—for bioplastics to decompose effectively (*Narancic et al., 2020*). Bioplastics that meet compostability standards, like ASTM D6400 or EN 13432, can be processed in these facilities. The resulting compost is then used to enrich soil in agriculture or landscaping, closing the organic waste loop (*Pesaranhajiabbas et al., 2022*). Compostable bioplastics should not be placed in backyard composting systems unless the manufacturer specifies, they are suitable for home composting. These bioplastics need specific conditions for proper breakdown (*Folino et al., 2020*). Some bioplastics, especially those made from starch or certain compostable polymers, are compatible with home composting systems. They break down naturally in backyard composting environments where food scraps and yard trimmings decompose. To ensure effective degradation, follow guidelines provided by the manufacturer or local composting authorities (*Santi, Elegir & Del Curto, 2020*). Home composting is a practical and eco-friendly option for managing bioplastics on a smaller scale. However, effective recycling and composting require robust waste management infrastructure, including separate collection systems, clear labelling, and consumer awareness. Collaboration among governments, waste management authorities, industry stakeholders, and consumers is crucial for developing comprehensive recycling and composting systems (*Eriksen et al., 2018*). Continued research and innovation in these areas will enhance the effectiveness and efficiency of bioplastic disposal options (*Hamid et al., 2022*).

# IMPACT ON BIODIVERSITY AND ECOSYSTEMS

## Investigation of the ecological implications of bioplastic production and waste

The production and waste management of bioplastics have ecological implications that need to be carefully considered. The investigation of some key ecological implications associated with bioplastic production and waste.

### Land use and deforestation

Bioplastics often rely on plant-based materials like corn, sugarcane, or soy, which require agricultural land. Expanding this land for feedstock cultivation can lead to deforestation, loss of natural habitats, and the creation of monocultures, disrupting biodiversity and

reducing carbon sequestration (*Piemonte & Gironi, 2011*). The water-intensive methods used for growing and processing these feedstocks can contribute to water scarcity and pollution from runoff and waste water, negatively impacting aquatic ecosystems (*Helmes et al., 2018*; *López-Pacheco et al., 2022*). When bioplastic production depends on fossil fuels, it contributes to greenhouse gas (GHG) emissions and climate change. To mitigate these effects, using renewable energy sources like solar or wind is essential (*Melchor-Martínez et al., 2022*). Additionally, the manufacturing and refinement of bioplastics can generate air pollution if not managed properly, and intensive feedstock farming can cause soil erosion, threatening soil fertility and ecosystem health (*Qin et al., 2021*). Improper disposal of bioplastics, especially when they are not recycled or composted, can result in pollution. Non-degrading bioplastics in landfills or natural environments contribute to plastic pollution, which endangers wildlife through ingestion or entanglement (*Baldera-Moreno et al., 2022*; *Faleti, 2022*). Incineration of bioplastics, if not properly controlled, can release harmful pollutants into the air. To address these challenges, it is important to use water-efficient irrigation and wastewater treatment, prioritize renewable energy and energy-efficient technologies, and implement soil conservation practices. Developing comprehensive waste management systems that emphasize recycling and composting, along with consumer education on responsible waste management, are crucial steps (*Carballo-Sánchez et al., 2022*; *Notaro, Elisabetta & Alessandro, 2022*). By considering these ecological implications and adopting sustainable practices throughout the life cycle of bioplastics, we can minimize their environmental impact and support a transition to a more sustainable, circular economy (*de Souza Vandenberghe et al., 2021*). Although bioplastics offer benefits by reducing reliance on fossil fuels, their production and disposal still pose significant environmental challenges, such as deforestation, habitat loss, and potential soil and water contamination (*López-Pacheco et al., 2022*; *Kaur et al., 2023*; *Koottatep et al., 2023*).

To address these issues, adopting sustainable sourcing practices and responsible land use is crucial. Best management practices in agriculture, like precision techniques and organic farming, can help reduce soil and water contamination (*Kharb & Saharan, 2022*). Inadequate waste management of bioplastics can lead to marine pollution, where they break down into harmful microplastics that threaten aquatic life and enter the food chain (*Herzke et al., 2021*). To prevent this, it is important to promote recycling and proper disposal methods to keep bioplastics out of water systems. Addressing these ecological concerns involves sustainable feedstock sourcing, responsible manufacturing, effective waste management, and strong regulations for bioplastics. Encouraging sustainable agricultural practices and integrated pest management can further reduce environmental impacts (*Winkler, Viers & Nicholas, 2017*). Developing and enforcing regulations and infrastructure for waste management, including recycling facilities and public education campaigns, supports the responsible disposal of bioplastics. Promoting the use of biodegradable and compostable bioplastics that meet established standards for degradation can also help minimize ecological impacts (*Watt et al., 2021*). These combined efforts will enhance the potential of bioplastics as sustainable alternatives to traditional plastics.

# MITIGATION STRATEGIES AND SUSTAINABLE PRACTICES

## Overview of strategies to minimize environmental impacts

To reduce the environmental impact of bioplastics, a comprehensive approach is needed throughout their life cycle. Sustainable sourcing involves using responsibly obtained feedstocks to avoid deforestation, protect biodiversity, and manage land properly (*López-Pacheco et al., 2022*). This includes using non-food sources like agricultural residues or algae to avoid competing with food supplies. Improving resource efficiency in bioplastic production means optimizing processes, reducing energy use, and minimizing waste (*Gbadeyan, Linganiso & Deenadayalu, 2023*). This can be achieved by adopting energy-efficient technologies, recycling production by-products, and creating closed-loop systems to reduce resource extraction (*Sidek et al., 2019*). A Circular Economy approach involves designing bioplastics to be reusable, recyclable, or compostable (*Beltran et al., 2021*). Ensuring compatibility with existing recycling systems and setting up dedicated collection systems can help keep bioplastics out of landfills, thus reducing waste and conserving resources (*Reichert et al., 2020*). Developing strong recycling and composting infrastructure includes expanding facilities that can handle various types of bioplastics and promoting consumer education (*Ferreira, Alves & Coelhoso, 2016*). Providing composting environments suited for bioplastics allows them to break down into valuable compost. Raising consumer awareness about the benefits of bioplastics and proper disposal practices is also crucial (*Ambrosio et al., 2021*). Clear labelling and educational campaigns help consumers recycle and compost effectively. Policy initiatives support sustainable bioplastic production and disposal by setting standards for biodegradability and compostability, encouraging eco-friendly innovations with financial incentives, and enforcing effective waste management practices (*Abe, Branciforti & Brienzo, 2021*). Collaboration among governments, industries, researchers, and NGOs can drive improvements in bioplastics, reducing their environmental footprint and enhancing their sustainability (*Boneberg et al., 2016*). By adopting these strategies, we can minimize the environmental impact of bioplastics and advance a more sustainable, circular approach to plastic production and waste management (*Maione, Lapko & Trucco, 2022*).

## Promising innovations in bioplastics technology

In recent years, innovations in bioplastics technology have focused on improving their environmental performance and expanding their uses. Researchers are developing biopolymer blends that mix different bioplastics or combine them with other materials to enhance their properties (*Ziani et al., 2023*). These blends include materials like PLA, PHA, starch-based polymers, and cellulose-based polymers, which come from renewable resources and are biodegradable (*Acquavia et al., 2021*). They are used in areas such as packaging, medical devices, and textiles. While the production of bioplastics is increasing, ongoing research aims to optimize their properties and processing methods to make them more suitable for industrial use and sustainable (*Sid et al., 2021*). Bioplastic blends can offer better durability, heat resistance, and flexibility while remaining environmentally friendly. For example, blending PLA with other polymers can improve mechanical

properties and broaden its applications (*Filho et al., 2022*). Adding bio-based and biodegradable materials, such as natural fibers, starch, or nanocellulose, further enhances bioplastics (*Jeremić et al., 2020*). These additives can strengthen the bioplastics, improve their barrier properties, and boost their biodegradability (*Youssef et al., 2019*). Additionally, biodegradable polymer composites reinforced with natural fibres like hemp or bamboo provide increased strength and lighter weight, making them useful in automotive, construction, and packaging industries as eco-friendly alternatives (*Shaikh, Yaqoob & Aggarwal, 2021*; *Thanu & Deepak, 2022*; *Dixon & Wilken, 2018*).

### Role of government regulations and industry initiatives

Government regulations and industry initiatives play a crucial role in shaping the future of bioplastics. These regulations set clear standards for bioplastic production, including environmental performance, biodegradability, compostability, and safety (*Melchor-Martínez et al., 2022*; *Vinci et al., 2021*). By enforcing these standards, regulations ensure consistency, build consumer trust, and provide manufacturers with clear guidelines. Governments can boost bioplastics innovation by offering grants, tax incentives, and funding for research and development (Single-Use Plastic Recycling Funding Opportunity Announcement Topic; *Tan, Tiwari & Ramakrishna, 2021*). This financial support helps advance technology, ease market entry, and speed up the commercialization of sustainable bioplastics. Extended Producer Responsibility (EPR) policies require manufacturers to manage the EOL of their products, encouraging designs that support recycling or composting and fostering better infrastructure (*Ambrosio et al., 2021*). Restrictions or bans on non-biodegradable or single-use plastics encourage the use of bioplastics as greener alternatives, reducing plastic pollution and promoting a circular economy (*Preka et al., 2022*; *Nielsen et al., 2023*). These measures drive innovation in sustainable packaging and increase the use of renewable resources. Industry associations complement these regulations with voluntary sustainability programs, offering best practices, environmental performance metrics, and resource efficiency guidelines (*Xu, Cheng & Liao, 2018*). Collaborative initiatives within the bioplastics sector bring together companies, universities, and research institutions to explore new feedstocks, improve manufacturing processes, and overcome technical challenges (*Batog et al., 2021*). These partnerships will accelerate the development of sustainable solutions.

## FUTURE OUTLOOK

To evaluate the potential of bioplastics in reducing plastic pollution, it is essential to consider various aspects. This includes examining technological advancements, infrastructure needs, market demand, consumer awareness, and the environmental impacts of bioplastics. Ongoing research and technological improvements are expected to enhance the properties, biodegradability, and recyclability of bioplastics, making them more effective substitutes for conventional plastics (*López-Pacheco et al., 2022*). Developing strong recycling and composting systems is crucial to fully utilize bioplastics. This requires investments from governments, industry leaders, and waste management sectors (*Miksch et al., 2022*). Increasing consumer awareness about the environmental

impact of plastic pollution and the benefits of bioplastics is also important for boosting market demand and encouraging businesses to adopt sustainable practices (*Kakadellis, Lee & Harris, 2022*). Assessing the overall sustainability of bioplastics involves understanding their complex environmental effects. Future improvements in feedstock sourcing, manufacturing processes, and waste management strategies are expected to reduce the environmental footprint of bioplastics (*Siracusa & Blanco, 2020*). Adopting a circular economy approach is key to sustainable bioplastic use, focusing on recycling, composting, and using renewable feedstocks (*Guzman-Puyol, Benítez & Heredia-Guerrero, 2021*). Standardizing and improving LCAs will provide clear insights into the environmental impacts of bioplastics, aiding stakeholders in making informed decisions (*Di Bartolo, Infurna & Dintcheva, 2021*). A comprehensive approach to sustainability and waste management is vital for addressing environmental challenges related to plastic pollution. Moving towards a circular economy emphasizes resource efficiency, waste reduction, and sustainable production and consumption (*Fytianos et al., 2021*). Technological innovations are crucial for improving waste management systems, developing sustainable materials and packaging, and enhancing resource efficiency (*Raheem et al., 2022*). Collaboration among stakeholders is essential for sharing knowledge, mobilizing resources, and achieving effective and sustainable waste management solutions (*Barakat et al., 2022*). Additionally, industry efforts focus on consumer education and market development (*Marquez et al., 2022*). Public awareness campaigns and certification programs help consumers understand the benefits of bioplastics, promote responsible consumption, and provide guidance on proper waste management (*Filho et al., 2022*). Collaborative efforts also support recycling and waste management infrastructure development (*Costa et al., 2023*). By aligning regulatory frameworks with industry initiatives, governments and stakeholders create a supportive environment for the sustainable growth and widespread adoption of bioplastics (*Cowan et al., 2021*). This integrated approach is essential for transitioning to a more environmentally friendly and circular economy, addressing global plastic pollution challenges (*Dou, Li & Lu, 2022*).

## CONCLUSION

Bioplastics show promise in fighting plastic pollution and promoting sustainability because they are made from renewable resources, can biodegrade or compost, and have a smaller carbon footprint compared to traditional plastics. However, to fully realize their benefits, it is important to assess their entire life cycle, from the sourcing of raw materials and production processes to waste management and environmental impact. To minimize negative ecological effects and avoid depleting resources, sustainable production and management practices are essential. Although bioplastics offer advantages like renewable sourcing and biodegradability, their environmental impact can vary based on factors such as the choice of feedstock, land use, energy consumption, and waste management practices. Maximizing the benefits of bioplastics requires a comprehensive approach that includes responsible sourcing, energy-efficient production, development of infrastructure, and consumer education. Successful integration into the economy involves a balanced approach that includes innovation, regulation, behaviour change, and collaboration across

different sectors. Addressing plastic pollution and other environmental challenges requires a broad framework for sustainability and waste management. This means considering the entire life cycle of products to find ways to reduce waste, use resources efficiently, and minimize environmental impacts. Collaboration among stakeholders is key to driving systemic change and implementing practices that promote sustainability. Investing in waste prevention, using sustainable materials, building infrastructure, and adopting circular economy principles are crucial for achieving long-term sustainability and ensuring a healthier environment for future generations.

### Funding
The authors received no funding for this work.

### Competing Interests
Ganesh Chandrakant Nikalje is an Academic Editor for PeerJ.

### Author Contributions
- Kushi Yadav conceived and designed the review outline, performed the literature survey, analyzed the data, prepared figures and/or tables, authored or reviewed drafts of the article, and approved the final draft.
- Ganesh Chandrakant Nikalje conceived and designed the outline of the review, performed the literature survey, analyzed the data, prepared figures and/or tables, authored or reviewed drafts of the article, and approved the final draft.

### Data Availability
This is a literature review.

### Supplemental Information
Supplemental information for this article can be found online at http://dx.doi.org/10.7717/peerj.18013#supplemental-information.

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
