# Peer review of "Comprehensive analysis of bioplastics: life cycle assessment, waste management, biodiversity impact, and sustainable mitigation strategies"

_PeerJ, doi:10.7717/peerj.18013_

## Round 0.1 · original submission · Major Revisions

Please prepare a carefully revised version of the manuscript. It would be nice if the authors address all queries/suggestions in their revised version.

**Language Note:** The review process has identified that the English language must be improved. PeerJ can provide language editing services - please contact us at [email protected] for pricing (be sure to provide your manuscript number and title). Alternatively, you should make your own arrangements to improve the language quality and provide details in your response letter. – PeerJ Staff

Reviewer 1 ·

Basic reporting

The literature review falls short of expectations, primarily due to a lack of substantive comparative analysis across the selected research articles. The article makes statements about the environmental impacts of bioplastics but fails to provide a meaningful comparison of factual data.

The central critique revolves around the article's failure to deliver on its promise of a comparative analysis. The review lacks the necessary depth to effectively contrast findings from different research articles, relying heavily on broad statements rather than concrete numerical comparisons.

To enhance the rigor of the review, incorporate specific quantitative data when comparing environmental impact indicators across research articles. This will strengthen the validity of the conclusions drawn. Consider factors such as methodological differences, study populations, or geographic locations that could contribute to diverse outcomes. Revise the literature review to provide a more cohesive narrative. Clearly articulate the common themes and methodologies observed in the existing research, emphasizing the gaps that the current study aims to address.


Expand the discussion section to delve into the practical implications of the comparative analysis. Discuss how the identified patterns can inform policy decisions, influence industry practices, and guide future research directions.

Experimental design

The review leans towards qualitative assessments, lacking the inclusion of specific numerical data to support claims. The absence of concrete numbers hinders the ability to draw meaningful conclusions about the relative ecological footprints of bioplastics.

While the literature review outlines common themes and methodologies, it falls short in providing a clear synthesis of the existing research. The lack of a cohesive narrative makes it challenging to follow the logical progression of ideas and understand the overall landscape of bioplastics research.

Validity of the findings

The discussion section does not sufficiently address the implications of the comparative analysis or explore potential reasons behind discrepancies in findings. There is a need for a more in-depth examination of the practical implications for policy and industry practices.

Additional comments

While the literature review attempts to shed light on the ecological footprint of bioplastics, it falls short of its potential due to the lack of a robust comparative analysis and concrete quantitative data. Addressing the outlined concerns and incorporating the suggested improvements will significantly strengthen the article, making it a more impactful contribution to the understanding of bioplastics' environmental impacts.

Annotated reviews are not available for download in order to protect the identity of reviewers who chose to remain anonymous.

Reviewer 2 ·

Basic reporting

1. Please improve the title; it conveys a similar message in two parts of a sentence
2. The Abstract section needs language (try using some scientific terminology) and basic structure improvements. Also, include future direction.
3. Please revise the manuscript carefully to improve the language. Rewrite the following sentences;
Line 43-47, L 76-78, L 100-101, L 132-134, L 165-172 (Break into smaller sentences), L 178-180, L204-207, L 211-213, L 216-219, L 233-235, L 289-291, L 344-347, L 413-415
4. Omit repetition of sentences
Line 48-49, L 135, L 435-437, L 534-537
5. Please clarify the meaning of the sentence " Bioplastics offer versatile applications and can
58 be engineered to possess..." (L 57-59)
6. L 170-172, what were the conditions? Feedstock? Which cyanobacteria?
7. Revise the subheading 6.1

Experimental design

1. This review needs to be restructured, starting from introducing bioplastics, their biological sources and synthesis, challenges in usage, adoption, and degradation, life cycle assessment, risk assessment and mitigation, conclusion and prospects.
2. Figures 1 and 2 are not very easy to understand; try making them attractive and self-explanatory
3. Try including 1 or 2 tables to give some recent information about the synthesis/advances/challenges/ of the bioplastics.
4. The LCA section is written poorly and needs major revisions.
5. Methods for bioplastic production section should be divided into subsections, highlighting each method's challenges. Different methods, yield, source, and challenges could also include a table.
6. Sections 6.3 and 6.4 seem the repetition, but they could be merged into the relevant sections
7. Section 7: Authors have already described the impacts of improper disposal of bioplastics on the environment at various places; try removing those sentences in other sections or try adding some new information on this subject in this section.

Validity of the findings

1. What authors did get from the Pubmed search? Results of survey methodology? Is there any figure or table to support the findings?

Reviewer 3 ·

Basic reporting

In manuscript “Unveiling the ecological footprint: Examining the environmental impacts of bioplastics” the authors discuss the dire problem of plastic pollution that is point of concern for environmental, aquatic, and terrestrial ecosystems. Life cycle assessment and environment impact assessment are of great importance to understand the short and long-term effects of bioplastics as alternatives. However, poor and redundant representation of data fails to capture the essence of such an important topic. On technical basis, publishing of manuscript in “Peer J” has not been recommended. Few points are mentioned to improve the quality of manuscript.

Experimental design

• Overall, the structure of manuscript is poorly organized, redundant, leading to a lack of coherence in presenting the information which greatly affects the reader’s ability to comprehend the presented information. Comprehensive restructuring of the manuscript to improve clarity and logical progression is recommended.
• Contents discussed in “bioplastic production methods” are irrelevant and mere redundancy of previously discussed contents.
• The literature review is insufficient in terms of depth and coverage. A more exhaustive literature review (in terms of quantitative data) related to progress in bioplastic production is recommended.

Validity of the findings

Clear mentioning of manuscript's novelty is missing.
• Manuscript has 3 different conclusions and respective future outlooks, so it is recommended to merge all these conclusions and future outlooks in single comprehensive heading.

Additional comments

• Careful checking of manuscript for word spacing, typing, and formatting mistakes is recommended.
• Survey methodology has to be written in past tense.
• Information provided at line 355-356 “breakdown organic materials in the absence of oxygen. Some bioplastics, including certain types of biopolyesters and biogas production” has to be rechecked.
• Unnecessary word capitalization is discouraged, check and correct in manuscript accordingly.
• Visual representation of figure requires improvement.
• Figure 2 does not provide sufficient information, so it is recommended to delete this figure.
• The manuscript does not contain a single table, which is the essence of review paper to convey important information. Addition of at least 2 tables is recommended.

---

## Round 0.2 · Minor Revisions

Looking forward to authors for the minor changes/corrections!

Reviewer 2 ·

Basic reporting

OK

Experimental design

OK

Validity of the findings

OK

Reviewer 3 ·

Basic reporting

• Authors are encouraged to prioritize clarity by simplifying complex sentences and using straightforward language. Avoid unnecessary jargon and metaphors.

Experimental design

• Description of survey methodology must be in past tense. Pay special attention to sentences at line 105-109 and 126-129.

Validity of the findings

• Authors are suggested to share the results of survey that they conduct.

---

## Round 0.3 · accepted · Accept

The authors have addressed all comments!